# Selenium and Episodic Memory: The Moderating Role of Apolipoprotein E ε4

**DOI:** 10.3390/nu17030595

**Published:** 2025-02-06

**Authors:** Shin Gyeom Kim, Musung Keum, Young Min Choe, Guk-Hee Suh, Boung Chul Lee, Hyun Soo Kim, Jun Hyung Lee, Jaeuk Hwang, Dahyun Yi, Jee Wook Kim

**Affiliations:** 1Department of Neuropsychiatry, Soonchunhyang University Bucheon Hospital, Bucheon 14584, Republic of Korea; redmensch@schmc.ac.kr; 2Department of Neuropsychiatry, Hallym University Dongtan Sacred Heart Hospital, Hwaseong 18450, Gyeonggi, Republic of Korea; ms8989@naver.com (M.K.); ymchoe@hallym.or.kr (Y.M.C.); suhgh@chol.com (G.-H.S.); 3Department of Psychiatry, Hallym University College of Medicine, Chuncheon 24252, Gangwon, Republic of Korea; leeboungchul@hallym.or.kr; 4Department of Neuropsychiatry, Hallym University Hangang Sacred Heart Hospital, Seoul 07247, Republic of Korea; 5Department of Laboratory Medicine, Hallym University Dongtan Sacred Heart Hospital, Hwaseong 18450, Gyeonggi, Republic of Korea; kimhyun@hallym.or.kr; 6Department of Laboratory Medicine, Green Cross Laboratories (GC Labs), Yongin 16924, Gyeonggi, Republic of Korea; hermes2014@gmail.com; 7Department of Psychiatry, Soonchunhyang University Hospital Seoul, Seoul 04401, Republic of Korea; hju75@schmc.ac.kr; 8Institute of Human Behavioral Medicine, Medical Research Center Seoul National University, Seoul 03080, Republic of Korea; dahyunyi@gmail.com

**Keywords:** selenium, cognition, APOE4, episodic memory, Alzheimer’s disease

## Abstract

**Background:** Selenium (Se), a vital trace element, plays a neuroprotective role by mitigating oxidative stress through selenoproteins and regulating metal balance. The apolipoprotein E ε4 allele (APOE4), a significant genetic risk factor for Alzheimer’s disease (AD), has been linked to reduced Se levels and weakened antioxidant capacity. This research explores the association between serum Se concentrations and cognitive performance, with an emphasis on how APOE4 status influences this relationship. **Methods:** This study included 196 older adults from community and memory clinic settings, who underwent assessments for episodic memory, global cognition, and non-memory functions using the Consortium to Establish a Registry for Alzheimer’s Disease (CERAD) neuropsychological battery, with serum selenium levels analyzed via inductively coupled plasma–mass spectrometry (ICP-MS) and APOE genotyping conducted to determine allele status. **Results:** Higher serum Se levels were associated with better episodic memory score (EMS) (B = 0.065, 95% CI = 0.020–0.110, *p* = 0.005) and CERAD total score (TS) (B = 0.119, 95% CI = 0.046–0.193, *p* = 0.002). However, the interaction between Se and APOE4 status significantly affected EMS (B = −0.074, 95% CI = −0.109 to −0.039, *p* < 0.001), with significant benefits observed in APOE4-negative participants. **Conclusions:** This study highlights the genotype-specific impact of Se on cognitive health, emphasizing the need for personalized nutritional interventions targeting Se levels, particularly for APOE4-negative individuals. Future research should further elucidate the mechanisms of Se’s effects and assess its therapeutic potential in aging populations.

## 1. Introduction

Selenium (Se) is a crucial trace element with significant importance for human health, especially in supporting neuroprotection [1,2,3]. Its neuroprotective effects are primarily mediated through reducing oxidative stress via selenoproteins [2,4], modulating calcium influx through ion channels [5], and exhibiting anti-inflammatory properties [6]. Notably, Se deficiency has been linked to Alzheimer’s disease (AD) [7,8,9], a disorder characterized by beta-amyloid (Aβ) plaques, neurofibrillary tangles, neuro-inflammation, and oxidative stress [10,11,12]. Aβ aggregation is considered one of the early pathological events in AD, contributing to synaptic dysfunction, neuronal toxicity, and microglial activation, which further amplifies neuro-inflammation [13]. Additionally, Aβ pathology has been shown to exacerbate tau hyperphosphorylation, leading to neurofibrillary tangle formation and progressive neurodegeneration [14]. Hyperphosphorylated tau disrupts intracellular transport and promotes synaptic instability, further accelerating cognitive decline [15]. These findings suggest that Se could enhance the brain’s antioxidant defenses, potentially mitigating AD-related cognitive impairment [16,17].

Apolipoprotein E (APOE) indirectly influences Se delivery to neurons and glial cells through its role in lipid metabolism and neuroprotection [18,19]. APOE modulates the secretion of selenoprotein P (SELENOP), a key Se transport protein, by interacting with its heparin-binding sites. This regulation affects the availability of SELENOP for uptake by neurons and glial cells [18], thereby influencing Se distribution in the brain. Furthermore, APOE’s role in lipid transport and neuronal health may influence the expression and functionality of receptors such as APOE receptor-2 (APOER2), also referred to as low-density lipoprotein receptor-related protein 8 (LRP8), which facilitate SELENOP uptake [19]. Therefore, APOE indirectly affects Se delivery to neural cells by regulating SELENOP secretion and modulating receptor-mediated uptake mechanisms. 

The APOE ε4 allele (APOE4), a major genetic risk factor for late-onset AD [20,21], is associated with earlier Aβ deposition [22,23] and increased oxidative stress [24,25], contributing to its significant influence on AD pathophysiology. Emerging evidence indicates that serum Se levels are lower in APOE4 carriers compared to non-carriers, suggesting genotype-specific differences in Se metabolism [26,27]. Given APOE’s critical role in regulating SELENOP-mediated Se delivery to neurons, these lower Se levels in APOE4 carriers may exacerbate deficiencies in antioxidant defenses, potentially accelerating oxidative damage and cognitive decline. This underscores the significance of exploring the interaction between Se and APOE4 in the context of AD, as it could guide targeted nutritional strategies to alleviate cognitive impairment in at-risk populations.

Beyond AD, Parkinson’s disease (PD) shares common mechanisms, including neuro-inflammation and oxidative stress. Orexin, a regulator of sleep–wake cycles, also plays a neuroprotective role, with studies showing that orexin receptors influence neuro-inflammation in AD [28] and orexin dysfunction contributes to PD progression [29]. Given these shared mechanisms, co-pathologies may also affect Se metabolism and cognitive decline, underscoring the need for a broader perspective on Se’s neuroprotective role.

Despite these insights, research on the relationship between Se levels, cognitive function, and the APOE genotype remains limited. To address this gap, this study examines the association between serum Se levels, episodic memory, and global cognition in older adults without dementia, with a particular focus on the moderating role of APOE4 status.

## 2. Materials and Methods

### 2.1. Participants

This study is part of the General Lifestyle and AD (GLAD) study, a prospective cohort study initiated in 2020. As of July 2022, the study enrolled 196 adults aged 65 to 90 years who were free of clinical dementia, including 83 cognitively normal (CN) individuals and 113 individuals with mild cognitive impairment (MCI). Recruitment was conducted through a dementia screening program at the memory clinic of Hallym University Dongtan Sacred Heart Hospital, Hwaseong, South Korea. To complement clinic-based recruitment, additional participants were identified through community outreach efforts, including recommendations from previous participants, their families, or acquaintances. Careful selection ensured that the cohort represented the general population, with a focus on capturing a broad spectrum of cognitive health in older adults. Participants in the CN group were those with a Clinical Dementia Rating (CDR) [30] score of 0 and no history of MCI or dementia. Individuals with MCI were identified based on established amnestic MCI criteria, requiring informant-confirmed memory concerns, measurable memory deficits, preserved overall cognitive function, independence in daily living activities, and no evidence of dementia. Memory impairment was assessed using age-, education-, and sex-adjusted z-scores, with scores below −1.0 on at least one of four episodic memory assessments from the Korean version of the Consortium to Establish a Registry for Alzheimer’s Disease (CERAD) neuropsychological battery [31,32]: word list memory, word list recall, word list recognition, and constructional recall [31,32,33]. All MCI participants had a CDR score of 0.5. Participants were excluded if they had significant psychiatric or neurological disorders, comorbidities affecting cognitive function, illiteracy, severe sensory deficits, difficulties with communication, or behavior affecting clinical evaluation, or if they were taking experimental medications.

### 2.2. Standard Protocol Approvals, Registrations, and Participants’ Consent

The study protocol received approval from the Institutional Review Board of Hallym University Dongtan Sacred Heart Hospital and was carried out in compliance with the latest guidelines of the Declaration of Helsinki. Informed consent was obtained from all participants or their legal representatives.

### 2.3. Clinical Assessments

Participants underwent comprehensive clinical evaluations conducted by trained psychiatrists in accordance with the GLAD study’s standardized assessment protocol, which includes the CERAD clinical and neuropsychological battery [31,32]. Licensed psychologists with expertise in geriatric populations administered the CERAD neuropsychological battery [33]. All clinical evaluations and diagnoses were based on a consensus approach involving psychiatrists and psychologists with expertise in dementia. The assessments included measures of episodic memory, non-memory cognitive domains, and overall cognitive performance. Episodic memory decline, a hallmark early symptom of Alzheimer’s disease [34,35,36,37,38,39], was quantified using the episodic memory score (EMS), calculated by summing scores from four tasks in the CERAD neuropsychological battery: word list memory, word list recall, word list recognition, and constructional recall. The non-memory score (NMS) was calculated by adding the scores from three non-memory tests: verbal fluency (executive function/attention/language), the modified Boston Naming Test (language), and constructional praxis (visuospatial and perceptual abilities) [40]. Global cognition was assessed using the CERAD total score (TS) [41], derived by summing the results of seven tests: verbal fluency, modified Boston Naming Test, word list memory, constructional praxis, word list recall, word list recognition, and constructional recall. Vascular risks were evaluated during structured interviews conducted by trained researchers with participants and their family members. The vascular risk score (VRS), expressed as a percentage, was calculated by summing the presence of conditions such as hypertension, diabetes mellitus, dyslipidemia, coronary heart disease, transient ischemic attack, and stroke [42]. Nutritional assessments were performed using the Mini-Nutritional Assessment (MNA) tool [43], a validated instrument for evaluating nutrition in older adults. The MNA considered factors such as recent reductions in food intake due to appetite loss, digestive issues, or chewing/swallowing difficulties, as well as dietary patterns, including protein and fruit/vegetable consumption. To ensure the reliability of the data, informants were interviewed when necessary [43].

### 2.4. Measuring Serum Levels of Selenium and Other Blood Biomarkers

Blood samples were collected in the morning (8–9 A.M.) via venipuncture and placed in trace element-free tubes to avoid contamination. Selenium concentrations were measured using an iCAP-RQ inductively coupled plasma–mass spectrometry (ICP-MS) (Thermo Fisher Scientific, Waltham, MA, USA). Calibration was performed using a multi-element standard solution (Part No. 5183-4688, Agilent Technologies, St. Clara, CA, USA), diluted in 1% (*v*/*v*) nitric acid and 0.5% (*v*/*v*) hydrochloric acid to construct four-point calibration curves, including a zero point. A blank, calibration standards, and quality control materials (low, medium, and high concentrations) were included with every batch of 20 samples to ensure precision and accuracy. The limit of blank (LOB), limit of detection (LOD), and limit of quantification (LOQ) for selenium were 0.17 µg/L, 0.39 µg/L, and 0.60 µg/L, respectively. Quality control materials showed excellent reproducibility, with coefficients of variation (CV) of 3.76% for low concentrations, 3.29% for medium concentrations, and 3.04% for high concentrations.

Serum albumin, glucose, HDL cholesterol, and LDL cholesterol were measured using the COBAS c702 analyzer with dedicated reagents supplied by Roche Diagnostics (Mannheim, Germany).

### 2.5. APOE4 Genotyping

Blood samples were obtained using vacutainer tubes containing EDTA as an anticoagulant. DNA was isolated with the QIAamp DSP DNA Blood Mini Kit, supported by the QIAcube HT System (QIAGEN, Hilden, Germany). Genotyping for APOE was conducted using the Seeplex ApoE ACE Genotyping Kit (Seegene, Seoul, Republic of Korea) on a ProFlex PCR system (Thermo Fisher Scientific, Waltham, MA, USA), following the provided instructions. After amplification, PCR products underwent analysis with the QIAxcel Advanced System (QIAGEN, Hilden, Germany), a capillary electrophoresis device, and genotypes were classified as ε2/ε2, ε2/ε3, ε2/ε4, ε3/ε3, ε3/ε4, or ε4/ε4 based on electrophoretic patterns and manufacturer guidelines. The presence of one or more ε4 alleles indicated APOE4 positivity.

### 2.6. Statistical Analysis

The association between serum Se levels and cognitive performance was evaluated using multiple linear regression analyses, treating selenium (as a continuous variable) as the independent variable and cognitive scores (EMS, NMS, and TS) as the dependent variables. Recognizing the potential impact of confounders on this relationship, factors such as age, sex, APOE4 status, education, clinical diagnosis, vascular risk factors, dietary habits (including protein and fruit or vegetable intake), and blood markers (albumin, glucose, HDL, and LDL cholesterol levels) were identified as covariates. Two stepwise models were applied: the first model adjusted for age, sex, APOE4 status, education, clinical diagnosis, and vascular risk factors, while the second model further included dietary protein and fruit or vegetable intake, as well as albumin, glucose, HDL, and LDL cholesterol levels.

To evaluate whether APOE4 positivity moderated the relationship between serum Se levels and cognitive decline, multiple linear regression analyses were performed with two-way interaction terms between Se levels and cognition included as independent variables. When significant interactions were observed, stratified analyses were conducted separately for APOE4-positive and APOE4-negative groups. Sensitivity analyses were conducted on individuals with cognitive impairment, given the reduced sensitivity of cognitive scores and the low APOE4 positivity rate in CN individuals (*n* = 15). Similar analyses were also carried out on participants without reduced food intake in the past 3 months to control for the possible effects of physical or mental health conditions on serum Se levels and cognitive performance. These were performed using SPSS Statistics software ver. 28 (IBM, Armonk, NY, USA).

## 3. Results

### 3.1. Participant Characteristics

The demographic and clinical characteristics of the study population, comprising 156 APOE4-negative and 40 APOE4-positive participants, are summarized in Table 1. All participants were non-demented, with 57% classified as having MCI and 43% as CN. Serum Se levels differed significantly between the two groups. In contrast, no significant differences were noted for any other demographic or clinical variables.

### 3.2. Association of the Serum Se Levels with Cognition

Serum Se level was significantly associated with EMS (B = 0.065, 95% confidence interval = 0.020 to 0.110, *p* = 0.005) and TS (B = 0.119, 95% confidence interval = 0.046 to 0.193, *p* = 0.002), but not with NMS (Table 2 and Figure 1A,B).

### 3.3. APOE4 Moderation of the Association Between the Serum Se Levels and Cognition

Serum Se levels interacted significantly with APOE4 positivity, impacting EMS (B = −0.074, 95% CI: −0.109 to −0.039, *p* < 0.001) and TS (B = −0.069, 95% CI: −0.128 to −0.010, *p* = 0.022), but showing no significant effect on NMS or the three non-memory tests (Table 3 and Appendix A). This indicates that APOE4 moderates the associations between serum Se levels and episodic memory as well as global cognition, without affecting non-memory cognitive domains or individual non-memory tests (verbal fluency for executive function/attention/language, modified Boston Naming Test for language, and constructional praxis for visuospatial and perceptual skills). In APOE4-negative participants, serum Se levels were significantly associated with EMS (B = 0.072, 95% CI: 0.025 to 0.119, *p* = 0.003) and TS (B = 0.125, 95% CI: 0.046 to 0.204, *p* = 0.002), while no significant associations were found in the APOE4-positive subgroup (Table 4 and Figure 1C–F).

### 3.4. Sensitivity Analyses

The results of the sensitivity analysis of participants with MCI and those without a decrease in food intake over the past 3 months were similar for the EMS and TS (Appendix A).

## 4. Discussion

This study demonstrates a significant association between Se levels and cognitive function, particularly episodic memory and global cognition, in older adults without dementia. The relationship was more pronounced in APOE4-negative individuals, highlighting the potential moderating effect of APOE4 status.

Our results align with prior studies that have documented Se’s protective effects on cognitive health. Several observational and interventional studies have demonstrated that higher serum Se levels or Se supplementation are associated with improved cognitive outcomes, particularly in episodic memory. For instance, several cross-sectional and prospective longitudinal studies showed that low serum Se deficiency correlates with cognitive impairment or decline, especially in aging populations [44,45,46]. Similarly, a randomized controlled pilot trial reported that Se supplementation improved cognitive performance in older adults with MCI [47]. These studies provide strong evidence supporting the neuroprotective role of Se, particularly through its ability to mitigate oxidative stress and inflammation in the brain.

Our findings reveal distinct differences in the association between Se levels and cognitive outcomes based on APOE4 status. In APOE4-negative individuals, higher Se levels were significantly associated with better episodic memory and global cognition, whereas no such association was observed in APOE4-positive individuals. This APOE genotype-specific effect may be partly explained by the significantly lower Se levels observed in APOE4-positive individuals compared to APOE4-negative individuals (Table 1). These findings suggest that Se’s neuroprotective effects are more robust in APOE4-negative individuals, potentially due to their relatively higher baseline Se levels, which enhance its antioxidant and neuroprotective mechanisms. Although prior studies have highlighted the protective role of Se in cognitive health, they have largely overlooked the impact of the APOE genotype. By identifying this critical distinction, our study underscores the necessity of considering genetic factors, such as APOE4 status, when evaluating the neuroprotective effects of Se. This approach can provide deeper insights into personalized strategies for preventing cognitive decline.

Se exerts its neuroprotective effects through multiple mechanisms that help explain our findings. As an essential micronutrient, Se is a key component of selenoproteins, such as glutathione peroxidase and thioredoxin reductase, which neutralize ROS and maintain redox balance in the brain [48]. These enzymes reduce oxidative stress [49], a major driver of neuronal damage and AD pathology [50]. By limiting oxidative damage, Se may help prevent the oxidative stress-induced aggregation of Aβ and hyperphosphorylation of tau [51,52], thereby preserving cognitive function. Beyond its antioxidant properties, Se plays a critical role in maintaining metal homeostasis, which is closely linked to oxidative stress in AD. Imbalances in metal ions such as copper, iron, and zinc contribute to neurodegeneration by promoting oxidative stress through Fenton reactions [53,54]. Se mitigates this effect by forming inert metal–selenium complexes, which are safely excreted, reducing metal-induced oxidative toxicity [55]. Collectively, these neuroprotective properties underscore Se’s potential to counteract oxidative stress and metal toxicity, key contributors to AD pathology, thereby potentially delaying or mitigating AD-related cognitive decline.

In this study, all participants were non-demented, with 57% classified as having MCI and 43% as CN. Given that MCI represents an early stage of cognitive impairment, identifying factors that may influence its progression is crucial. Emerging evidence suggests that selenium’s neuroprotective mechanisms may be particularly relevant in MCI, a transitional stage between normal aging and dementia [56]. Oxidative stress and neuro-inflammation play a central role in MCI pathogenesis, contributing to early synaptic dysfunction and neuronal damage [57,58]. Se, through its role in antioxidant defense and metal homeostasis, may help mitigate these effects by reducing ROS, modulating neuro-inflammatory pathways, and preventing amyloid and tau pathology [48,49,50,51,52,53,54]. Furthermore, studies suggest a potential link between Se deficiency and MCI. One study found that specific Se species in cerebrospinal fluid may predict MCI progression to AD [59]. Another study reported that elevated selenoprotein P levels in serum and cerebrospinal fluid were associated with an increased risk of MCI conversion to dementia [60]. These findings highlight the need for further research to clarify the role of Se in MCI and its progression.

The interaction between Se and APOE4 status suggests genotype-specific differences in Se metabolism and its impact on cognitive health. APOE4 carriers exhibit lower serum Se levels than APOE4-negative individuals (Table 1), likely due to differences in Se transport, metabolism, or utilization rather than a simple deficiency threshold. APOE4 is known to influence lipid and metal homeostasis, which may affect Se incorporation into neuroprotective selenoproteins [18,19]. Impaired Se transport across the blood–brain barrier or reduced selenoprotein activity in APOE4 carriers could weaken antioxidant defenses, increasing their vulnerability to oxidative stress-related cognitive decline. In contrast, APOE4-negative individuals may utilize Se more efficiently, resulting in a stronger association between higher Se levels and better cognitive outcomes. Rather than implying a specific Se threshold for cognitive benefits, our findings highlight the importance of considering the APOE genotype when evaluating Se’s role in cognition. This perspective may help guide genotype-specific nutritional strategies to support cognitive health.

Our study has several limitations that should be considered when interpreting the findings. First, its cross-sectional design limits our ability to establish causality between serum Se levels and cognitive function. Longitudinal studies are needed to confirm whether higher Se levels directly protect against cognitive decline over time or are merely correlated with better cognitive health. Second, while we controlled for potential confounders, including vascular risks, nutritional biomarkers, and dietary patterns, unmeasured variables such as other genetic polymorphisms beyond APOE or dietary Se intake could still have influenced our findings. However, sensitivity analyses provided reassurance: participants whose food intake did not decline over the past three months exhibited similar results (Appendix A), suggesting that dietary Se deficiency was unlikely to explain the observed associations. Additionally, additional regression analyses that included blood nutritional biomarkers and dietary patterns as covariates (Table 2 and Table 4; Appendix A) yielded consistent results, further mitigating concerns about confounding from dietary factors. Third, the relatively small number of APOE4-positive participants limits the scope of the subgroup analyses and decreases the statistical power needed to detect subtle associations between serum Se levels and cognitive outcomes in this subgroup. Future research should involve larger and more diverse cohorts to confirm and refine these observations. Fourth, this study did not include key AD-related biomarkers, such as Aβ and tau, or APOE4-associated metabolic markers. Incorporating these biomarkers in future investigations would provide valuable insights into how serum Se levels relate to AD pathophysiology and cognitive decline, especially in APOE-negative older adults. Addressing these limitations will contribute to a more comprehensive understanding of the role of selenium in cognitive health and its interactions with APOE genotypes.

## 5. Conclusions

Our study demonstrates a significant association between Se levels and cognitive function, particularly episodic memory and global cognition, in older adults without clinical dementia. This relationship was more pronounced in individuals without the APOE4 allele, suggesting a potential protective role of Se in cognitive health, influenced by genetic factors such as APOE status. These findings emphasize the need to consider genetic variability when exploring the neuroprotective effects of Se and its role in mitigating cognitive decline. Future research should aim to clarify the mechanisms underlying Se’s effects and assess the potential of Se-based interventions in preserving cognitive function, particularly in aging populations.

## Figures and Tables

**Figure 1 nutrients-17-00595-f001:**
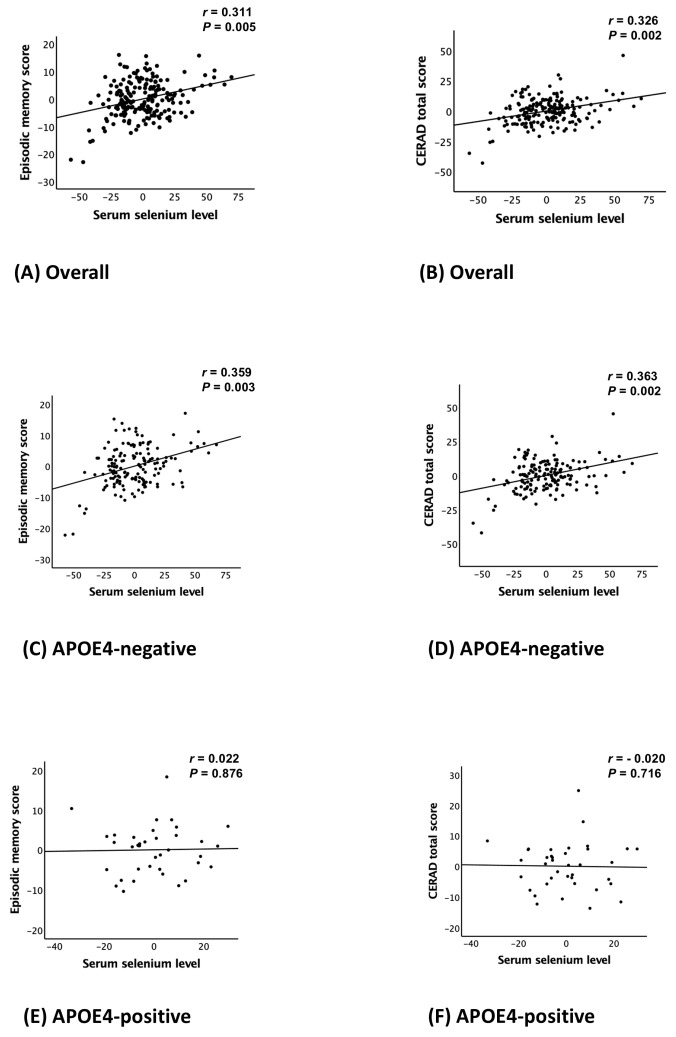
Partial regression plot for the association between the serum Se levels and episodic memory score (**A**,**C**,**E**) and CERAD total score (**B**,**D**,**F**) in non-demented older adults according to the APOE4 status. Abbreviations: Se, selenium; CERAD, the Consortium to Establish a Registry for Alzheimer’s Disease; APOE4, apolipoprotein ε4 allele. Multiple linear regression analyses were performed after adjusting for all confounders.

**Table 1 nutrients-17-00595-t001:** Demographic and clinical characteristics of the participants according to APOE4 status.

	Overall	APOE4-Negative	APOE4-Positive	*p*
*n*	196	156	40	
Age, y	72.65 (5.95)	72.95 (5.96)	71.50 (5.86)	0.170 ^a^
Female, *n* (%)	138 (70.41)	106 (67.95)	32 (80.00)	0.136 ^b^
Education, y	9.62 (4.51)	9.61 (4.55)	9.68 (4.38)	0.934 ^a^
MCI, *n* (%)	113 (57.65)	88 (56.41)	25 (62.50)	0.487 ^b^
VRS, %	23.98 (18.58)	23.93 (19.14)	24.17 (16.43)	0.943 ^a^
MMSE	25.58 (3.45)	25.52 (3.46)	25.83 (3.43)	0.618
Dietary pattern including food types				
Protein, *n* (%)				0.410 ^b^
High	27 (13.78)	19 (12.18)	8 (20.00)	
Moderate	74 (37.76)	59 (37.82)	15 (37.50)	
Low	95 (48.47)	78 (50.00)	17 (42.50)	
Fruit or vegetables, *n* (%)				0.795 ^b^
High	119 (60.71)	62 (39.74)	15 (37.50)	
Low	77 (39.29)	94 (60.26)	25 (62.50)	
Decrease in food intake over the past three months				1.00 ^c^
No, *n* (%)	182 (92.86)	145 (92.95)	37 (92.50)	
Yes, *n* (%)	14 (7.14)	37 (23.72)	3 (7.50)	
Serum nutritional markers				
Se, ug/L	110.06 (21.62)	111.88 (22.51)	103.23 (16.38)	0.024 ^a^
Albumin, g/dL	4.57 (0.26)	4.57 (0.26)	4.60 (0.25)	0.465 ^a^
Glucose, fasting, mg/dL	108.15 (19.94)	108.46 (21.02)	106.87 (14.87)	0.660 ^a^
HDL cholesterol, mg/dL	54.64 (12.96)	54.51 (12.89)	55.21 (13.38)	0.765 ^a^
LDL cholesterol, mg/dL	96.41 (33.82)	96.10 (35.42)	97.68 (26.64)	0.796 ^a^
Cognition				
Memory score				
EMS	35.10 (9.48)	35.17 (9.47)	34.83 (9.67)	0.840 ^a^
Non-memory score				
NMS	34.25 (6.62)	34.06 (6.92)	35.00 (5.26)	0.423 ^a^
Global cognition				
TS	69.98 (15.61)	70.00 (16.15)	69.90 (13.52)	0.971 ^a^

Abbreviations: APOE4, apolipoprotein E ε4 allele; MCI, mild cognitive impairment; VRS, vascular risk score; MMSE, mini-mental state examination; Se, selenium; HDL, high-density lipoprotein; LDL, low-density lipoprotein; EMS, episodic memory score; NMS, non-memory score; TS, total score of the Consortium to Establish a Registry for Alzheimer’s Disease. Data are expressed as mean (standard deviation), unless otherwise indicated. ^a^ By Student’s *t* test; ^b^ by chi-square test; ^c^ by Fisher’s exact test.

**Table 2 nutrients-17-00595-t002:** Results of the multiple linear regression analyses of the association between the serum Se levels and cognitive decline.

	B	95% CI	*p*
EMS			
Model 1	0.061	0.019 to 0.104	0.005
Model 2	0.065	0.020 to 0.110	0.005
NMS			
Model 1	0.028	−0.005 to 0.061	0.096
Model 2	0.033	−0.003 to 0.068	0.069
TS			
Model 1	0.117	0.048 to 0.186	<0.001
Model 2	0.119	0.046 to 0.193	0.002

Abbreviations: Se, selenium; APOE4, apolipoprotein E ε4 allele; EMS, episodic memory score; TS, total score of the Consortium to Establish a Registry for Alzheimer’s Disease; VRS, vascular risk score. The first model included age, sex, APOE4, VRS, education, and clinical diagnosis as covariates; the second model included those covariates plus protein intake, fruit/vegetable, albumin, fasting glucose, and HDL or LDL cholesterol.

**Table 3 nutrients-17-00595-t003:** Results of multiple linear regression analyses that included interaction terms for the association between serum Se levels and APOE4 positivity in predicting cognitive decline.

	B	95% CI	*p*
EMS			
Se levels	0.079	0.035 to 0.123	<0.001
APOE4 positivity	6.655	2.915 to 10.395	<0.001
Se levels × APOE4 positivity	−0.074	−0.109 to −0.039	<0.001
NMS			
Se levels	0.035	−0.001 to 0.070	0.059
APOE4 positivity	1.724	−1.346 to 4.794	0.269
Se levels × APOE4 positivity	−0.009	−0.038 to 0.020	0.546
TS			
Se levels	0.132	0.059 to 0.205	<0.001
APOE4 positivity	6.182	−0.078 to 12.441	0.053
Se levels × APOE4 positivity	−0.069	−0.128 to −0.010	0.022

Abbreviations: Se, selenium; APOE4, apolipoprotein ε4 allele; EMS, episodic memory score; NMS, non-memory score; TS, total score of the Consortium to Establish a Registry for Alzheimer’s Disease. To explore the moderating effects of APOE4 positivity on the associations between serum selenium level and cognition, i.e., EMS, NMS, and TS, multiple linear regression analyses were performed including two-way interaction terms between serum selenium level and cognition as additional independent variables.

**Table 4 nutrients-17-00595-t004:** Results of the multiple linear regression analyses of the association between the serum Se levels and cognitive decline according to APOE4 subgroup.

	B	95% CI	*p*
EMS			
APOE4-negative			
Model 1	0.065	0.021 to 0.109	0.004
Model 2	0.072	0.025 to 0.119	0.003
APOE4-positive			
Model 1	0.031	−0.109 to 0.171	0.656
Model 2	−0.014	−0.196 to 0.168	0.876
TS			
APOE4-negative			
Model 1	0.126	0.052 to 0.199	<0.001
Model 2	0.125	0.046 to 0.204	0.002
APOE4-positive			
Model 1	0.040	−0.155 to 0.235	0.681
Model 2	−0.042	−0.274 to 0.191	0.716

Abbreviations: Se, selenium; APOE4, apolipoprotein E ε4 allele; EMS, episodic memory score; TS, total score of the Consortium to Establish a Registry for Alzheimer’s Disease; VRS, vascular risk score. The first model included age, sex, APOE4, VRS, education, and clinical diagnosis as covariates; the second model included those covariates plus protein intake, fruit/vegetable, albumin, fasting glucose, and HDL or LDL cholesterol.

## Data Availability

The study data are not freely accessible because the IRB of the Hallym University Dongtan Sacred Heart Hospital prevents public sharing of such data for privacy reasons. However, the data are available on reasonable request after IRB approval. Requests for data access can be submitted to an independent administrative coordinator by e-mail (yoon4645@gmail.com).

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
