# Peer review of "Selenium and Episodic Memory: The Moderating Role of Apolipoprotein E ε4"

_nutrients, 2025, doi:10.3390/nu17030595_

Round 1

Reviewer 1 Report

Comments and Suggestions for Authors

Kim et al elaborate on the significance of selenium in the context of episodic memory. Authors indicate that ApoE negative have higher levels of selenium, moreover in ApoE negative, higher levels of selenium improved cognitive outcomes. Low levels of selenium in ApoE positive were linked with deviated neuroprotection and likely increased oxidative stress and cognitive deterioration. I have the following comments regarding this work:

1. Regarding the goal of the research, authors should extensively elaborate on the pathogenesis of Alzheimer's disease in the introduction and discuss factors impacting cognitive deterioration. It would be valuable to acknowledge the factors in the context of possible impact in other neurodegenerative diseases due to possible co-pathologies - Ref. (A) Orexin receptors exert a neuroprotective effect in Alzheimer’s disease (AD) via heterodimerization with GPR103. Sci Rep 5, 12584 (2015). https://doi.org/10.1038/srep12584, (B) Role of orexin in pathogenesis of neurodegenerative parkinsonisms. Neurol Neurochir Pol. 2023;57(4):335-343. doi:10.5603/PJNNS.a2023.0044

2. The mechanisms linking selenium and oxidative stress should be more extensively described.

3. Authors should justify using MMSE instead of MoCA/ACE-III in the cognitive examination.

4. In the discussion authors acknowledged the role of selenium among patients with MCISimilarly, a r. This andomized

controlled pilot trial reported314
that Se supplementation improved cognitive performance in older adults315
witThis feature should be more extensively discussed in the context of patomechanisms.

1Se levels and improved cognitive outcomes in APOE4

-358

negative individuals.AP OE4 carriers, exhibit significantly lower serum Se

352
levels ( compared to APOE4-negative individuals

Author Response

­­Reviewer 1

Comments and Suggestions for Authors

Kim et al elaborate on the significance of selenium in the context of episodic memory. Authors indicate that ApoE negative have higher levels of selenium, moreover in ApoE negative, higher levels of selenium improved cognitive outcomes. Low levels of selenium in ApoE positive were linked with deviated neuroprotection and likely increased oxidative stress and cognitive deterioration. I have the following comments regarding this work:

  1. Regarding the goal of the research, authors should extensively elaborate on the pathogenesis of Alzheimer's disease in the introduction and discuss factors impacting cognitive deterioration. It would be valuable to acknowledge the factors in the context of possible impact in other neurodegenerative diseases due to possible co-pathologies - Ref. (A) Orexin receptors exert a neuroprotective effect in Alzheimer’s disease (AD) via heterodimerization with GPR103. Sci Rep 5, 12584 (2015). https://doi.org/10.1038/srep12584, (B) Role of orexin in pathogenesis of neurodegenerative parkinsonisms. Neurol Neurochir Pol. 2023;57(4):335-343. doi:10.5603/PJNNS.a2023.0044

=> We appreciate your suggestion to elaborate on the pathogenesis of Alzheimer’s disease (AD) and its broader implications for cognitive deterioration. In response, we have revised the Introduction to provide a more comprehensive discussion by incorporating neuroinflammation and oxidative stress as key contributors to AD, alongside Aβ aggregation and tau hyperphosphorylation. Additionally, we have acknowledged the shared mechanisms between AD and Parkinson’s disease (PD), including orexin dysfunction, neuroinflammation, and oxidative stress, to highlight the potential impact of co-pathologies on Se metabolism and cognitive decline. These revisions ensure a broader and more integrative perspective on selenium’s neuroprotective role while maintaining clarity and conciseness. We appreciate the reviewer’s valuable feedback, which has strengthened the contextual foundation of our study (Introduction: p3, lines 84-91; p4, lines 110-115; References: 10-17, 28,29).

  1. The mechanisms linking selenium and oxidative stress should be more extensively described.

=> We appreciate your suggestion to further elaborate on the mechanisms linking Se and oxidative stress. In response, we have expanded our discussion on selenoproteins’ role in redox regulation, highlighting how glutathione peroxidase and thioredoxin reductase neutralize ROS and maintain cellular redox balance. Additionally, we have clarified how Se reduces oxidative stress-induced Aβ aggregation and tau hyperphosphorylation, key contributors to Alzheimer’s disease pathology. Furthermore, we have included its role in metal homeostasis, where Se mitigates oxidative stress by forming inert complexes with toxic metal ions. These revisions provide a more comprehensive explanation of Se’s neuroprotective mechanisms. We appreciate the reviewer’s valuable feedback, which has strengthened the mechanistic framework of our study (Discussion: p10-11, lines 279-291).

  1. Authors should justify using MMSE instead of MoCA/ACE-III in the cognitive examination.

=> We appreciate the reviewer’s comment regarding the choice of MMSE over MoCA/ACE-III in the cognitive examination. In our study, we utilized the Consortium to Establish a Registry for Alzheimer’s Disease (CERAD) neuropsychological battery, which includes the MMSE as part of its standardized assessment. Given that our cognitive evaluation was based on this comprehensive neuropsychological battery, MMSE was selected to ensure consistency with the established testing protocol. We thank the reviewer for this observation and hope this clarification addresses their concern (Methods: p5, lines 152-155; References: 31-33).

  1. In the discussion authors acknowledged the role of selenium among patients with MCI

This feature should be more extensively discussed in the context of patomechanisms.

=> Thank you for your insightful comment. We have expanded our discussion to further elaborate on selenium’s neuroprotective mechanisms in MCI. Emerging evidence suggests that selenium may mitigate oxidative stress and neuroinflammation, key contributors to MCI pathogenesis, by reducing ROS, modulating neuroinflammatory pathways, and preventing amyloid and tau pathology. Additionally, studies indicate a potential link between selenium levels and MCI progression, with specific Se species in cerebrospinal fluid associated with conversion to AD. In response to your suggestion, we have incorporated these points and relevant references to enhance the discussion. We appreciate your feedback, which has strengthened our manuscript (Discussion: p11, lines 292-304, References: 56-60).

Reviewer 2 Report

Comments and Suggestions for Authors

This is an interesting cross-sectional epidemiological study that provides further insight into the potential role of selenium in age-related cognitive dysfunction. Importantly, not only do the authors provide further evidence that higher serum selenium levels are associated with better memory in aging populations but they also show that this effect appears to be restricted to APOE4 minus individuals. Although it is known that APOE can modulate selenium levels through its effects on SELENOP and LRP8, how the APOE4 genotype affects these functions does not appear to be well studied. Hopefully, this manuscript can stimulate further research on this important question. There are a few points however that require clarification.

1.        In several points in the manuscript that authors state that the subjects in the study did not have dementia. However, according to Table 1, 57% had MCI so slightly more than half were not cognitively normal. This should be made clear throughout the text.

2.        Table 1 lists 196 participants overall with 138 of those female. But the columns for APOE4 negative and positive list only 58 females total. Could the authors please clarify.

3.        Figure 1: What does 0 indicate on the x-axis? Why weren’t actual selenium values used?

4.        The argument in the Discussion that the APOE genotype specific effect on the association between cognitive function and serum selenium levels is related to the lower selenium levels in APOE4 individuals does not make sense to me. Are the authors implying that there is a certain threshold level of selenium which needs to be exceeded in order to see better cognitive function? Please clarify.

Author Response

Reviewer 2

Comments and Suggestions for Authors

This is an interesting cross-sectional epidemiological study that provides further insight into the potential role of selenium in age-related cognitive dysfunction. Importantly, not only do the authors provide further evidence that higher serum selenium levels are associated with better memory in aging populations but they also show that this effect appears to be restricted to APOE4 minus individuals. Although it is known that APOE can modulate selenium levels through its effects on SELENOP and LRP8, how the APOE4 genotype affects these functions does not appear to be well studied. Hopefully, this manuscript can stimulate further research on this important question. There are a few points however that require clarification.

  1. In several points in the manuscript that authors state that the subjects in the study did not have dementia. However, according to Table 1, 57% had MCI so slightly more than half were not cognitively normal. This should be made clear throughout the text.

=> Thank you for your valuable comment. As you suggested, we have clarified the cognitive status of the study participants to ensure accuracy. Specifically, we have revised Section 3.1. Participants to explicitly state that while none of the participants had dementia, 57% had MCI, meaning that slightly more than half were not CN. Furthermore, we have added a paragraph in the Discussion section to further elaborate on the potential effects of selenium in MCI. This addition highlights selenium’s role in mitigating oxidative stress and neuroinflammation, key factors in MCI pathogenesis, and discusses recent findings suggesting a link between selenium levels and MCI progression. We appreciate your feedback, which has helped enhance both the clarity and depth of our manuscript (Results: p8, lines 227,228; Discussion: p11, lines 292-304; References: 56-60).

  1. Table 1 lists 196 participants overall with 138 of those female. But the columns for APOE4 negative and positive list only 58 females total. Could the authors please clarify.

=> We appreciate your invaluable comments and apologize for the error in Table 1. You are correct in noting the discrepancy in the reported number of female participants. The correct numbers are 106 females in the APOE4-negative group and 32 females in the APOE4-positive group, which sum to the total of 138 female participants. We have corrected this in Table 1 to accurately reflect the study population. Thank you for bringing this to our attention, and we appreciate your careful review, which has helped improve the accuracy of our manuscript (Table 1).

  1. Figure 1: What does 0 indicate on the x-axis? Why weren’t actual selenium values used?

=> Thank you for your question. In Figure 1, the x-axis represents the residualized values of serum selenium levels after adjusting for all confounders in the multiple linear regression model. The value 0 on the x-axis corresponds to the expected selenium level after accounting for confounders, meaning that positive and negative values reflect deviations from this adjusted mean. We used partial regression plots instead of raw selenium values to visualize the independent association between selenium and episodic memory score while controlling for covariates. This approach helps to isolate the unique contribution of selenium to cognitive performance, minimizing the influence of confounding variables. We appreciate your careful review and hope this clarification addresses your concern.

  1. The argument in the Discussion that the APOE genotype specific effect on the association between cognitive function and serum selenium levels is related to the lower selenium levels in APOE4 individuals does not make sense to me. Are the authors implying that there is a certain threshold level of selenium which needs to be exceeded in order to see better cognitive function? Please clarify.

=> Thank you for your insightful comment. We appreciate the opportunity to clarify our argument regarding the APOE genotype-specific effect on the association between cognitive function and serum selenium levels. Our intention was not to suggest that there is a strict selenium threshold that must be exceeded to observe cognitive benefits. Rather, we propose that the observed differences between APOE4 carriers and non-carriers may be due to differences in selenium metabolism, transport, and incorporation into selenoproteins, which play a crucial role in neuroprotection. APOE4 carriers exhibit lower serum selenium levels, but more importantly, they may have altered selenium homeostasis, including impaired transport across the blood-brain barrier or reduced incorporation into essential selenoproteins. This could limit the neuroprotective effects of selenium, making them more susceptible to oxidative stress-related cognitive decline. In contrast, in APOE4-negative individuals, where selenium metabolism may be more efficient, higher selenium levels could contribute more effectively to cognitive resilience. To better reflect this interpretation, we have revised the Discussion section to emphasize that the APOE4-specific effect likely results from differences in selenium metabolism rather than a simple threshold requirement for cognitive benefits. We appreciate your feedback, as it has helped us refine the clarity and interpretation of our findings (Discussion: p11, 305-316).

Round 2

Reviewer 1 Report

Comments and Suggestions for Authors

I do not have further comments.